# Mechanistic analysis and kinetic profiling of Soai's asymmetric autocatalysis for pyridyl and pyrimidyl substrates

Patrick Möhler[1,3], Gloria Betzenbichler[1,3], Laura Huber[1], Alexander F. Siegle[1] & Oliver Trapp [1,2] ✉

Nonlinear effects in chemical reactions, coupled with amplifying catalysis, can lead to remarkable phenomena like spontaneous symmetry breaking, central to the origin of biological homochirality. Soai's asymmetric autocatalysis is a prototypical reaction for this, where the enantiomeric excess of the product alcohol is amplified during alkylation of pyridyl and pyrimidyl carbaldehydes by diisopropylzinc. However, the complex equilibria and elusive intermediates make the mechanism difficult to clarify. Here we unravel the intricate dynamics of this reaction by in situ high-resolution mass spectrometry, kinetic analysis, and reaction profile simulations. We identify for both the pyrimidyl and the pyridyl systems transient hemiacetalate isopropyl zinc complexes, formed by the addition of the alcoholate product to the aldehyde, as key catalytic intermediates. These diastereomeric complexes enable dual stereo-control, explaining the observed enantioselectivity. Our analysis confirms the structures of all intermediates and validates the autocatalytic cycle, offering insights into how substituent and structural variations influence reaction performance. This understanding guides the design of new, efficient asymmetric autocatalytic systems.

In Soai's asymmetric autocatalysis[1] pyrimidyl and pyridyl[2,3] carbaldehydes are alkylated by diisopropyl zinc, which is facilitated by the product alcohol. This unique reaction is so far the only known example combining autocatalysis and self-amplification. Due to this uniqueness, the Soai reaction is often considered as a model reaction for the origin of homochirality in nature[4–7]. 2-alkynyl substituted pyrimidine systems proved to be highly efficient and robust substrates in regard to self-amplification, as an *ee* as low as $10^{-5}\%$ of the autocatalytic alcohol is enhanced to more than 99.5% in only three reaction cycles (Fig. 1a)[8,9].

The mechanistic elucidation of the Soai reaction and the origin of the strong positive nonlinear effect (+NLE)[10,11] have been the subject of intensive research for the last three decades. It seems evident that the origin of the NLE is attributed to the formation of homo- and heterochiral dimers of the zinc alkoxide, which are initially formed from the alcohol additive and then alter as the end product of the reaction. However, the question of the catalytically active species is still under discussion[12,13].

In early mechanistic considerations, a homochiral dimeric zinc alkoxide was proposed as the catalytically active structure, while the heterochiral dimer is catalytically inactive[14–25]. Subsequent studies provided insights into the aggregation of pyrimidyl zinc alkoxides and the idea of the dimeric zinc alkoxide catalyst was displaced by a homochiral tetrameric zinc alkoxide as catalytically active structure[26–30]. Especially the square-macrocycle-square (SMS) tetramer consisting of two dimeric zinc alkoxide units connected via Zn-N coordination, as proposed by Blackmond, was discussed as plausible oligomeric structure with substrate binding sites (Fig. 1b)[27]. The connectivity of the SMS tetramer shows a significant distinction from proposed cubic tetramers of zinc alkoxides as described by Noyori and coworkers[31]. Experimental

---

[1]Department of Chemistry, Ludwig-Maximilians-University Munich, Munich, Germany. [2]Max Planck Institute for Astronomy, Heidelberg, Germany. [3]These authors contributed equally: Patrick Möhler, Gloria Betzenbichler. ✉e-mail: oliver.trapp@cup.uni-muenchen.de

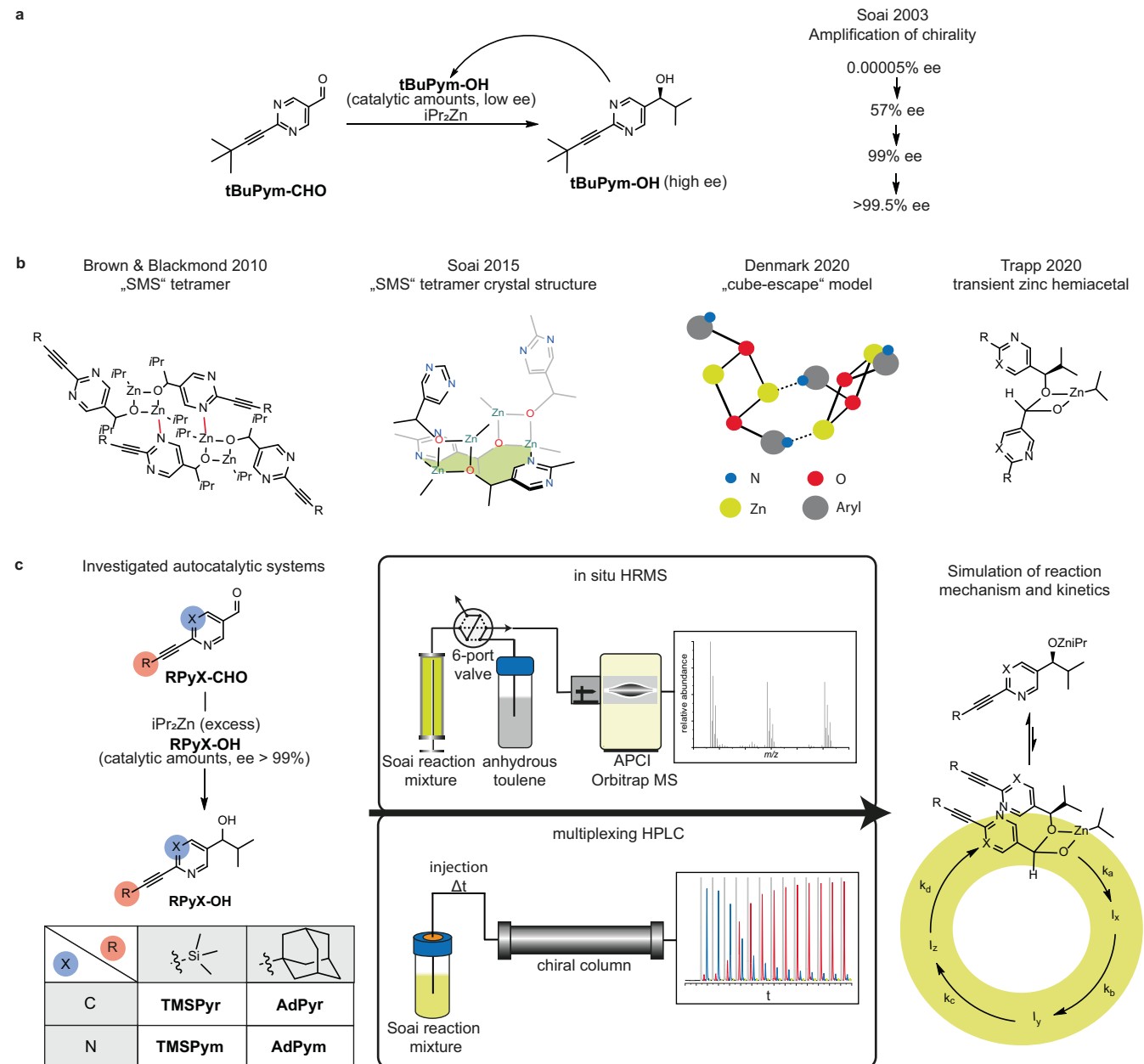

**Fig. 1 | Soai's asymmetric autocatalysis and overview of mechanistic investigations. a** Highly efficient self-amplifying autocatalytic reaction of 2-(tert-butylethynyl)pyrimidine-5-carbaldehyde (**tBuPym-CHO**) with iPr₂Zn, yielding >99.5% ee within four reaction cycles discovered by Soai et al.[9]. **b** Milestones of mechanistic investigations on the Soai reaction. **c** Overview of the four autocatalytic systems investigated in this work: pyridyl systems (**TMSPyr** and **AdPyr**) and pyrimidyl systems (**TMSPym** and **AdPym**). Intermediates were characterized by in situ HRMS reaction profiling and kinetic analysis was conducted by multiplexing HPLC. Experimental data was used for the reaction network analysis and kinetic simulation of the reaction mechanism.

evidence for the existence of these stable SMS-tetramers was provided by NMR spectroscopy and single-crystal X-ray diffraction (Fig. 1b)[27,32,33]. In an in depth study by Denmark and coworkers, the structural influence of the aromatic core, the alkyl groups of the carbinol carbon and the zinc alkylating agent on aggregation was investigated. The necessity of one heterocyclic nitrogen atom and isopropyl groups on the carbinol and zinc for autocatalytic behaviour was pointed out[34,35]. The activation of dialkylzinc reagents by coordinating functional groups and ligands, e.g., β-amino alcohols, is well known[36–39]. As proposed, these structural features enable the formation of non-cubic zinc alkoxide tetramers ("cube-escape"), which are, in contrast to cubic tetramers, potentially catalytically active (Fig. 1b)[34,35].

Further findings were published by Blackmond and Brown in 2012, who reported the observation of a hemiacetal intermediate during the

Soai reaction by NMR[40]. It was found that the amount of the hemiacetal increases with higher initial aldehyde concentration and especially with decreasing temperatures. Therefore, a remarkable relationship between the hemiacetal intermediate and kinetic peculiarities of the Soai reaction, such as a higher order rate dependency in aldehyde concentration and an inversed temperature effect, can be derived from these results[27,40].

In 2020, a mechanistic pathway was published by our group that proceeds via the transient chiral zinc hemiacetalate as the catalyst for the asymmetric alkylation by activating the reactants through coordination to this chiral hemiacetal (Fig. 1b)[41,42]. The proposed mechanism was derived from reaction intermediates identified by in situ HRMS measurements and kinetic data, which disclosed a second-order rate dependency in aldehyde concentration. Here we present comprehensive

results from detailed mechanistic and kinetic investigations, which give fundamental insights into the mechanism of the Soai reaction. To address open questions regarding the very different reaction behaviour of pyridyl and pyrimidyl derivatives, detailed experiments were performed. This allows the evaluation of the structure-reactivity relationship of these derivatives in a catalytic mechanistic model based on in situ formed diastereomeric hemiacetal-ligand-zinc complexes.

## Results and discussion

We investigated four autocatalytic systems in detail under consistent reaction conditions. Tetramethylsilane- (TMS) as well as adamantyl- (Ad) acetylene residues combined with either a pyridyl (Pyr) or a pyrimidyl (Pym) entity were chosen as autocatalytic reaction systems (Fig. 1c). The TMS- and Ad-acetylene residues on the heterocycle reportedly enable an efficient autocatalytic behaviour and complement previous results from our group[41]. It should be pointed out, that the **AdPyr** system constitutes an autocatalytic system which has not been investigated in previous works.

Reaction intermediates of each system were examined by in situ high-resolution mass spectrometric (HRMS) reaction profiling. Systematic kinetic measurements by time-resolved reaction progress analysis of the reactants and chiral products were conducted by chiral multiplexing HPLC[43,44]. Kinetic data as well as the detected intermediates were further used as a basis for the reaction network analysis and the kinetic simulation of the reaction mechanism (Fig. 1c).

### In situ high-resolution mass spectrometric reaction profiling

The identification and temporal tracking of reaction intermediates was enabled by pulsed injection of the Soai reaction mixture into an Orbitrap mass spectrometer under inert ($N_2$ as protective gas) and mild ionization conditions using atmospheric pressure chemical ionization ($T = 150\,°C$, $N_2$). The reaction conditions were selected to ensure that the reaction proceeded under homogeneous conditions. In situ HRMS measurements were carried out with the four investigated autocatalytic systems. Several reaction intermediates were observed which give further insights into the mechanism of the Soai reaction (Fig. 2a).

In addition to the protonated monomeric zinc alkoxide $I_1$, the dimeric zinc alkoxide $I_2$ was detected for the TMS-substituted systems. Dimeric species also occurred with a cleaved isopropyl group or with a hydroxy group coordinating to a zinc atom replacing an isopropyl group (Supplementary Information Figs. 10 and 11). Higher aggregates of the zinc alkoxide, such as tetramers, were not detected under these reaction conditions for any of the four autocatalytic systems.

During these measurements several hemiacetal intermediates were identified. The hemiacetal $I_3$ resulting from the precursor aldehyde and the alcohol was detected in the protonated form and with a cleaved hydroxy group. Interestingly, the exact molecular mass of the protonated ester $I_4$ was detected, which according to Rotunno et al. can be formed via hydride transfer from the zinc hemiacetal to a precursor aldehyde to form the ester and a primary zinc alkoxide in a Claisen-Tishchenko reaction[45].

For all four investigated systems the hemiacetal resulting from an aldehyde and a zinc alkoxide $I_5$ was observed. An analogous zinc hemiacetal intermediate was previously reported by our group during the autocatalytic alkylation of *tert*-butylethynyl substituted pyrimidine-5-carbaldehydes (*t*BuPym, Fig. 1a)[41]. The formation of these hemiacetals is the key in the extraordinary efficient asymmetric alkyl transfer in the Soai reaction. By reaction of the chiral product alcohol with the aldehyde, an additional stereocenter is formed, which results in diastereomeric ligand for the dialkylzinc reagent. This means that one of the diastereomers is preferentially formed and thus a double stereocontrol is enabled, as has already been observed in other catalysts with a pronounced +NLE, e.g., the Sharpless epoxidation[10].

The conducted measurements enabled the observation of missing reaction intermediates, which complement the autocatalytic hemiacetal pathway. For both pyridine-based systems **TMSPyr** and **AdPyr** the molecular mass of an intermediate, which corresponds to the complex of the zinc hemiacetal catalyst and an aldehyde substrate $I_6$ was detected. Furthermore, the intermediate complex $I_7$ of the zinc hemiacetal and a diisopropyl zinc was observed. These two intermediates $I_6$ and $I_7$ constitute the alkylation step of the proposed autocatalytic cycle, in which the coordination to the hemiacetal activates the reactants and enables the asymmetric alkyl transfer from the sterically less demanding side (Fig. 2b). The resulting complex of the hemiacetal and the zinc alkoxide was observed in form of intermediate $I_8$. Thus, experimental evidence is provided for the existence of key intermediates of the alkylation step in the hemiacetal-mediated autocatalytic cycle under reaction conditions.

Furthermore, the pulsed injection with intermittent flushing with anhydrous toluene allowed tracking the temporal evolution and change of the reaction intermediates. Similar reaction profiles of these intermediates were observed for all four autocatalytic systems (Supplementary Figs. 1–3). The profiles of the catalytically active hemiacetal show a parabolic curve, in which the zinc hemiacetalate is formed during the early stages of the reaction when alcohol formation is slow. The concentration of the zinc hemiacetalate complex increases until an apex is reached at the inflection point of the s-shaped sigmoidal kinetic profile of the formed product alcohol and then eventually depletes. As shown for the **TMSPyr** system, the alkylation rate accelerates with increasing amounts of the zinc hemiacetalate until the apex of the zinc hemiacetal is reached after approximately 10 min (Fig. 2c). The maximum of this profile coincides with the phase of fast alcohol formation and aldehyde consumption (autocatalytic s-shaped sigmoidal profile), which is consistent with an increasing reaction rate at higher catalyst concentration. With decreasing aldehyde concentration, the amount of the hemiacetal depletes. Interestingly, a similar curve was also observed for the transient hemiacetal during tracking via NMR at $0\,°C$ by Blackmond and Brown[46]. Given that we observe similar results for five different autocatalytic systems during in situ HRMS measurements, it seems evident that the autocatalytic pathway for pyridine- and pyrimidine-based substrates proceeds in a similar mechanistic pathway, which involves the zinc hemiacetal as catalytically active species. To further elucidate the influence of different structural elements on the autocatalytic reactivity, extensive kinetic investigations were conducted.

### Experimentally derived kinetics

The progress of the reactions was examined by performing multiplexing HPLC in the flow-injection mode. Here, an injection of the reaction solution at constant time intervals on a chiral column allowed real-time analysis and quantification of all reactants and the product enantiomers at each point of the reaction progress. It has to be pointed out that by this technique only the stable reactants and products, such as free alcohols, are determined since alcoholates and zinc complexes are protonated upon entering the separation column by the mobile phase, while the in situ HRMS reaction profiling technique allows to monitor reactive intermediates under inert conditions. The first injection took place 30 s after initiation of the reaction. For the reaction graphs, the concentrations of the aldehyde and the alcohol enantiomers were calculated for each individual injection by means of a calibration line and plotted against time (Reaction profiles for all measured concentrations can be found in the Supplementary Information).

To determine the reaction orders of the reactants, alcohol and aldehyde concentrations were varied independently. The $i$Pr$_2$Zn concentration was kept at a constant 40 mM as it has already been observed in previous experiments that the Soai reaction follows a zeroth-order rate dependency in $i$Pr$_2$Zn concentration[27,34,41]. Kinetic

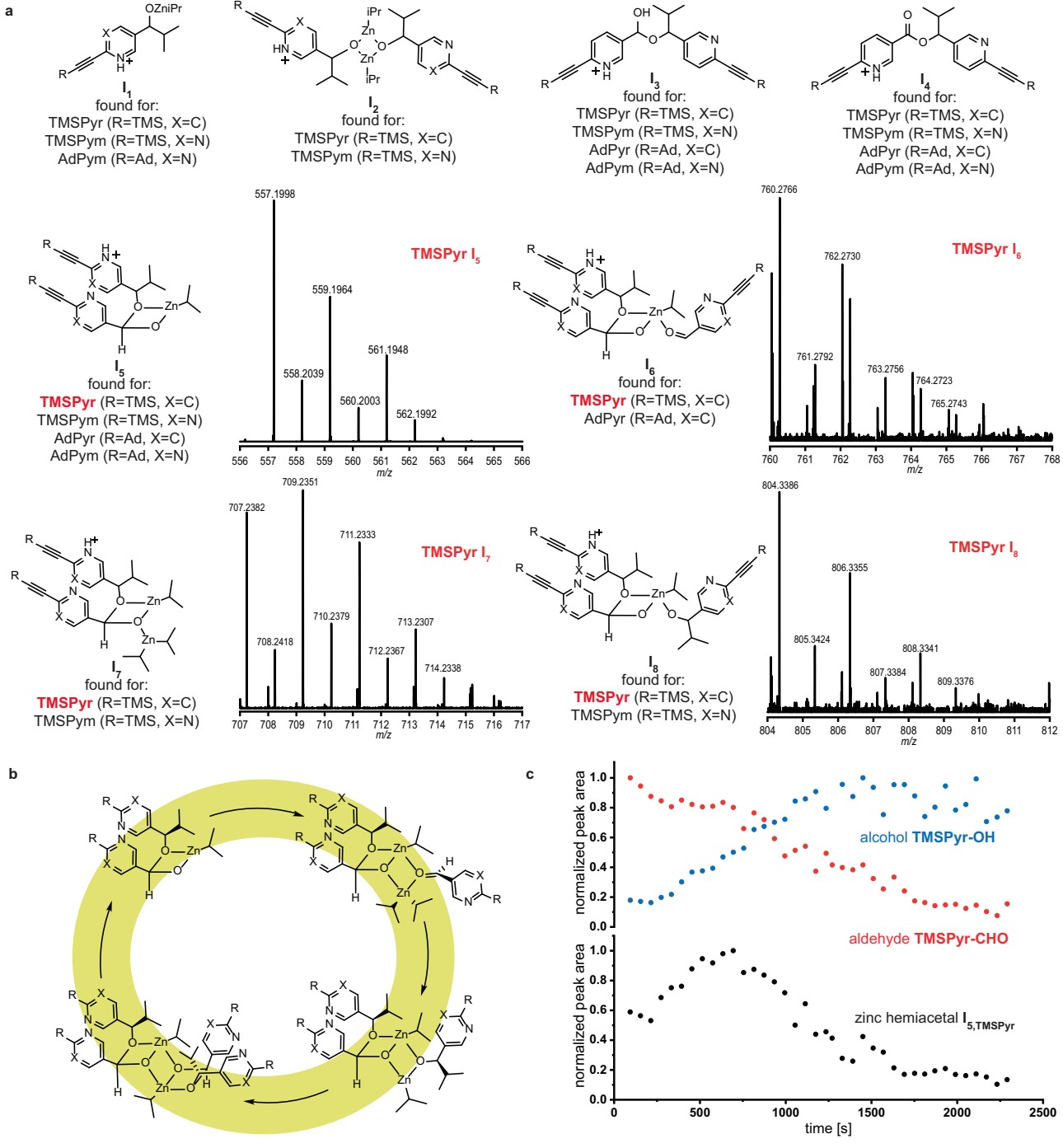

**Fig. 2 | In situ mass spectrometric identification and characterization of intermediates in Soai's asymmetric autocatalysis. a** Detected intermediates by in situ reaction HRMS profiling for the four investigated autocatalytic systems. Intermediates of the **TMSPyr** system represented in the autocatalytic cycle are shown with the characteristic isotope pattern of their molecular mass spectrum.

**b** Autocatalytic cycle of the Soai reaction as proposed by Trapp et al.[41]. **c** Normalized peak areas of the aldehyde **TMSPyr-CHO**, the alcohol **TMSPyr-OH** (top) and the zinc hemiacetalate complex **I₅** (bottom) plotted against time. Reaction conditions: 30.0 mM **TMSPyr-CHO**, 1.5 mM **TMSPyr-OH** (ee > 99%) and 40 mM iPr₂Zn in anhydrous toluene at r.t.

measurements were performed at initial aldehyde concentrations between 10 and 65 mM, adjusted to the system investigated. The initial concentrations of the enantiopure alcohol additive were varied between 0.7 and 5.3 mM. In this concentration range the characteristic autocatalytic sigmoidal reaction profiles are observable (Fig. 3). In general, it is clearly observed that the autocatalytic reaction proceeds significantly faster with increasing initial aldehyde and alcohol concentrations, indicating a reaction order greater than zero for both, pyridine and pyrimidine systems. This tendency can be seen in the comparison of low and high initial aldehyde concentrations at constant initial alcohol concentrations (Fig. 3). By double logarithmic plotting of initial velocity ($v_0$) versus concentration, the reaction order for the aldehyde and for the alcohol was determined for each system.

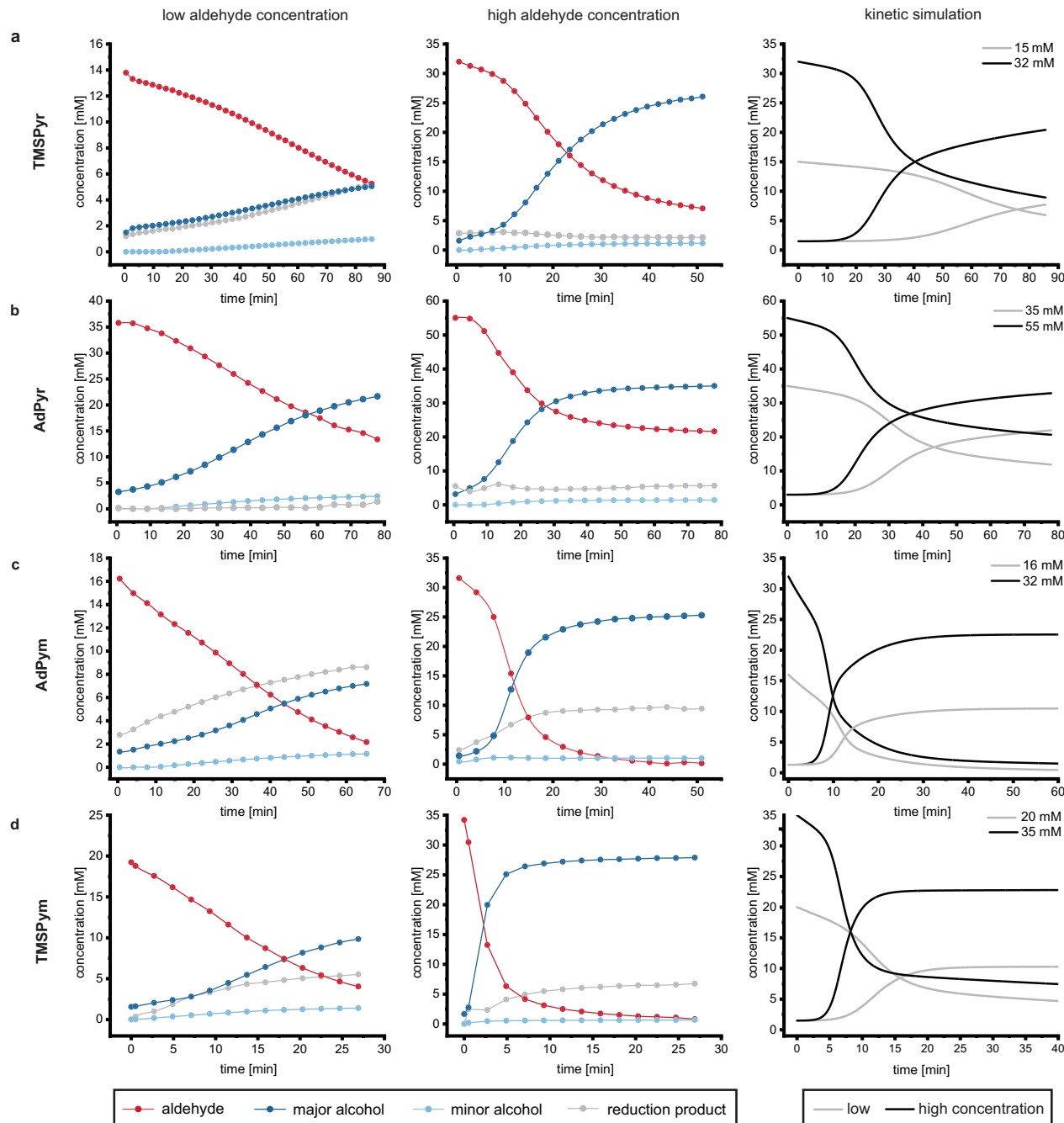

**Fig. 3 | Reaction progress of Soai's asymmetric autocatalysis of the four systems determined by chiral multiplexing concentrations (FIA)−HPLC at low and high initial aldehyde concentrations.** The right panels show the simulated kinetic profiles. **a** Variation of the initial concentrations of **TMSPyr-CHO** (15 and 32 mM) adding **TMSPyr-OH** (1.5 mM) and iPr₂Zn (40 mM). **b** Variation of the initial concentrations of **AdPyr-CHO** (35 and 55 mM) using **AdPyr-OH** (3.0 mM) and iPr₂Zn (40 mM). **c** Variation of the initial concentrations of **TMSPym-CHO** (20 and 35 mM) using **TMSPym-OH** (1.5 mM) and iPr₂Zn (40 mM). **d** Variation of the initial concentrations of **AdPym-CHO** (16 and 32 mM) using **AdPym-OH** (1.3 mM) and iPr₂Zn (40 mM).

In previous studies, reaction orders of 0[34], 1.6[27] and 1.9[41] were determined for the aldehyde. Here, it was found that when accounting for the side reaction, the reduction of the aldehyde to the primary alcohol, a reaction order of 2 is consistently obtained, regardless of the system considered. At low concentrations of aldehyde or alcohol, the side reaction can even proceed faster than the autocatalysis itself (Fig. 3, grey curves). The reduction of aldehydes to the primary alcohol is commonly observed during dialkylzinc alkylations[31]. To obtain valid reaction orders it is therefore necessary to not only monitor the aldehyde consumption, but also to simultaneously track the formation of the autocatalytic product and the reduction product. Here, FIA-HPLC-MS monitoring offers the unique advantage of a definite identification and tracking of the three possible reaction products, the two alkylated enantiomers and the reduced achiral side product. We assume that the differences of previous reported aldehyde reaction orders and those obtained here originate from the mandatory incorporation of the side reaction into kinetic considerations.

In agreement with previous publications, a first-order rate dependency in alcohol concentration was determined (Table 1)[27,34,41]. This further suggests that the mechanism of all four systems proceeds via a similar pathway, as the influence of aldehyde and alcohol concentration is similar for all substrates.

**Table 1 | Calculated reaction orders for the aldehyde and the alcohol from experimental data for the TMSPyr, AdPyr, AdPym and TMSPym systems**

| System | Reaction order Aldehyde RCHO | Reaction order Alcohol ROH |
|--------|------------------------------|----------------------------|
| TMSPyr | 2.01 ± 0.23 | 1.06 ± 0.09 |
| AdPyr | 2.03 ± 0.12 | 1.02 ± 0.21 |
| TMSPym | 2.02 ± 0.16 | 0.98 ± 0.02 |
| AdPym | 2.00 ± 0.11 | 0.97 ± 0.19 |

A comparison of the four systems shows that the pyrimidines react faster than the pyridine analogues. The electron-poor pyrimidine heterocycle increases electrophilicity of the aldehyde, favouring hemiacetal formation and thus the formation of catalytically active species. This distinct kinetic behavior of pyridine and pyrimidine autocatalytic systems displays the consistency of the structure-reactivity relationship with a mechanistic pathway involving a transient hemiacetal as the catalyst.

Furthermore, the kinetic data also disclosed the influence of the alkynyl residue on the autocatalytic reactivity. For the investigated systems, the TMS residues exhibit a faster reaction progress in comparison to the adamantyl analogues, regardless of whether they are connected to a pyrimidine or pyridine ring. This might be explained by the fact that, while an adamantyl group enhances solubility in toluene, the steric demand and electronic properties of the substituent also play a significant role with regard to autocatalytic behavior.

Comparing the two pyrimidine systems **AdPym** and **TMSPym** with the **tBuPym** previously investigated by our group at comparable concentrations[41], the latter is the fastest system for the enantioselective autocatalysis of the Soai reaction. This results in a descending order in autocatalytic reactivity of **tBuPym** > **TMSPym** > **AdPym** > **TMSPyr** > **AdPyr**, where the structural influences explained can be seen again at a glance. Therefore, the influence of the heterocycle on autocatalytic reactivity is more pronounced compared to the influence of the substituent on the alkynyl residue as seen in the comparison between the **AdPym** and **TMSPyr** system.

Temperature-dependent dynamic HPLC (DHPLC) experiments were conducted in order to determine the hemiacetal activation barriers for the four investigated aldehydes with isopropanol[47]. It was observed that the equilibrium between aldehyde and hemiacetal shifts increasingly towards the hemiacetal with decreasing temperatures, which essentially gives an explanation for the inversed temperature effect of the reaction kinetics as the catalytically active species is preferentially formed at lower temperatures[40]. Pyridine-based systems show an approximately 10 kJ/mol higher activation barrier for the hemiacetal formation (**TMSPyr**: 94.6 kJ/mol, **AdPyr**: 95.5 kJ/mol, **TMSPym**: 85.9 kJ/mol, **AdPym**: 86.4 kJ/mol). The substituent on the alkynyl group however, shows a significantly weaker influence as the Ad-substituted systems have an increased activation barrier of below 1 kJ/mol compared to their TMS analogues. The determined activation barriers in the model system are in excellent agreement with the observed autocatalytic reactivity in the Soai reaction. Systems with lower activation barriers for the hemiacetal formation exhibit higher autocatalytic reactivity.

## Kinetic modelling

Kinetic data was evaluated employing a reaction network model consisting of 18 ordinary differential equations (ODEs) for the autocatalytic mechanism (Supplementary Information Eqs. 5.2.1–5.2.18). The microscopic reversibility was considered in the equilibria of the formation of the dimeric alcoxides **2** and the hemiacetalate **3**. Therefore, equilibria constants were introduced to determine the backward reaction rates in these steps. Considering the experimental observation that the intermediates in the autocatalytic cycle are formed at low concentrations and immediately consumed by the successive steps, we neglected equilibria and the backward reactions because of the high conversion rates in these steps. This assumption is also based on the finding that the reverse reaction of the final dimeric alkoxide is not noticeably altered in the presence of diisopropyl zinc, which suggests, that the corresponding equilibria constants are high in the autocatalytic reaction cycle. Additionally, the reductive side reaction of the substrate aldehydes was incorporated into the model (Supplementary Information Eq. 5.2.19, 19 ODEs in total). The reaction rate of the reductive side reaction was independently determined from the reaction profiles by using the overall reaction rate law of the asymmetric autocatalysis using the here determined reaction orders and a kinetic equation describing the reduction. Implementation of the differential equations into a software program enabled the calculation of kinetic reaction profiles and reaction rate constants for every step of the autocatalytic mechanism (Fig. 4). For this purpose, a multidimensional equidistant parameter range for each variable in the ODE system was defined to cover all combinations of rate constants. This is computationally more demanding compared to the attempt to fit single parameters to the experimental data, however, it covers all possible scenarios. Approximately 24 million kinetic reaction profiles with all reactants, intermediates and enantiomer products are calculated in each iteration step using the experimental reaction conditions. The calculated kinetic parameters were refined by comparison of calculated and experimentally determined kinetic profiles with varying starting concentrations (Fig. 3). Here, the obtained reaction rates should be discussed exemplary for the autocatalytic (R)-cycle, which are the same for the (S)-cycle (mirror image).

Kinetic evaluation for the four different systems revealed, with $1.5 \times 10^2$ M$^{-1}$s$^{-1}$ a fast and structurally independent formation of the isopropylzinc alkoxides **(R)-1** from the alcohols in agreement with previous reports[48]. It is in equilibrium with its homo- **(R,R)-2** and heterochiral **(R,S)-2** dimers, which are diastereomeric to each other. For the dimerization of the zinc alkoxide, the equilibrium is generally shifted towards the dimers, but to a much greater extent for the pyrimidine-based alkoxides ($k_2$ & $k_3$). As the rapid formation of homochiral **(R,R)-2** and heterochiral **(R,S)-2** dimers is the underlying prerequisite for the development of equilibria, which lead to a self-amplification in the Soai reaction. This can be an explanation for a generally increased efficiency of the pyrimidine autocatalytic systems.

The formation of the zinc hemiacetal catalyst **(R,R)-3** from the zinc alkoxide **(R)-1** and a substrate aldehyde is the slowest and therefore the rate-determining step of the Soai reaction ($k_4$). This step is an order of magnitude faster for the two pyrimidine-based substrates than for the pyridine analogues, which essentially results in increased reactivity and shorter induction period in the autocatalytic reaction (Fig. 3). These values are also reflected in the kinetic HPLC experiments, where slower reaction progress was observed for the pyridine than pyrimidine systems.

Further, the coordination of a diisopropyl zinc molecule and a substrate aldehyde to the zinc hemiacetalate catalyst occurs fast ($k_5$), which is subsequently followed by the slower alkyl transfer ($k_6$). Ongoing from the resulting complex **(R,R,R)-5**, another aldehyde can coordinate fast ($k_7$) to give **(R,R,R,R)-6**, the dimeric form of **(R,R)-4**. The autocatalytic cycle is eventually closed by the slow proceeding dissociation into two monomeric zinc hemiacetals ($k_8$), while the newly formed hemiacetal **(R,R)-3** starts a second autocatalytic cycle.

In conclusion, the results presented here provide fundamental insights into the reaction mechanism of the Soai reaction. Through in situ high-resolution mass spectrometric reaction profiling of intermediates and detailed kinetic investigations—both experimental and via comprehensive reaction mechanism simulations—strong evidence is provided for a hemiacetal-catalyzed pathway. In particular, the identification of key intermediates in the autocatalytic cycle and the observation of second-order kinetics with respect to aldehyde concentration across all investigated systems corroborate the proposed mechanism.

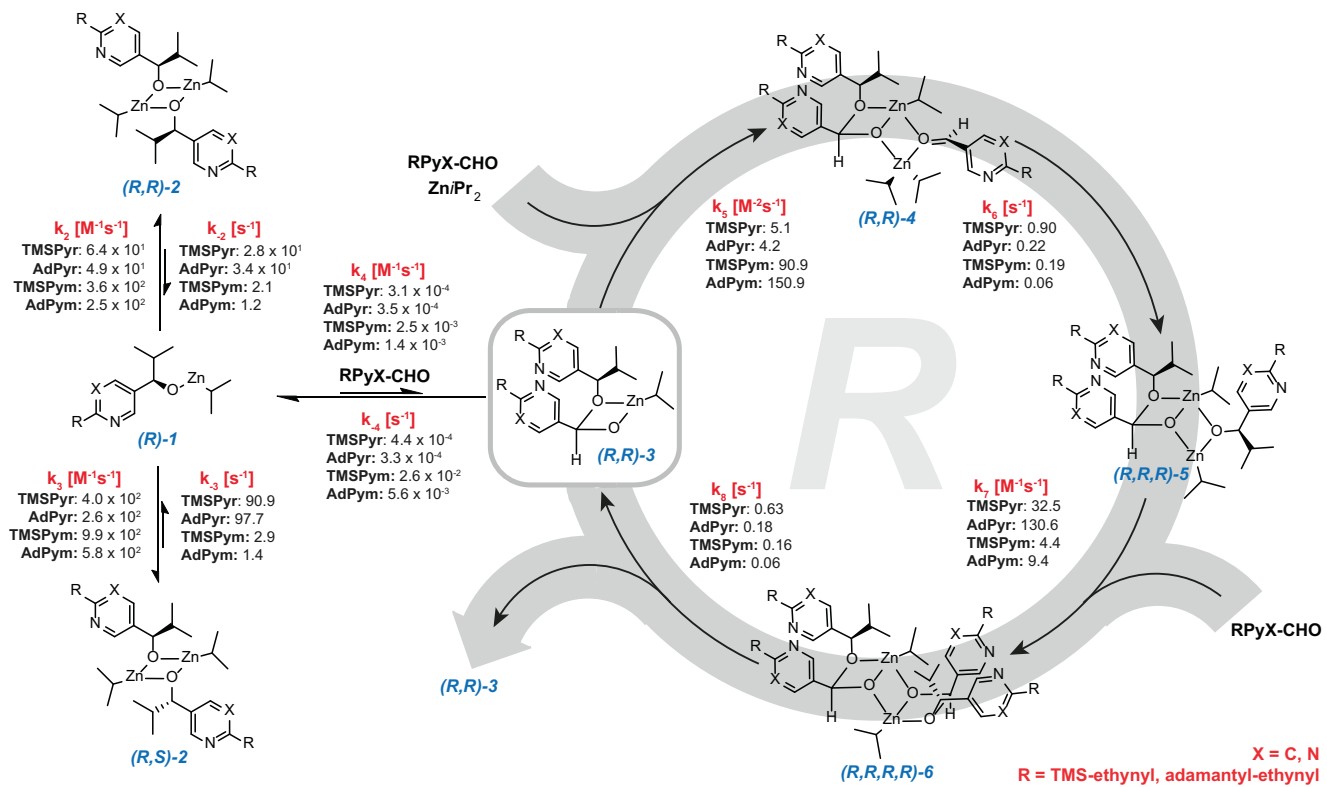

**Fig. 4 | Representation of the autocatalytic reaction mechanism of the Soai reaction for pyridine- and pyrimidine-based substrates.** The calculated rate constants are given in $M^{-1}s^{-1}$ for $k_2$, $k_3$, $k_4$, $k_5$, and $k_7$, in $s^{-1}$ for $k_6$, $k_8$, $k_{-2}$, $k_{-3}$ and $k_{-4}$.

The findings also reveal a clear structure-reactivity relationship between pyridine- and pyrimidine-based systems. In both systems, hemiacetals are formed via the reaction of the product alcohol with the aldehyde, existing as diastereomers due to the creation of an additional stereocenter.

These insights are particularly noteworthy because the underlying mechanism provides a fundamentally new approach for designing advanced autocatalytic reactions. Building on this work, new asymmetric autocatalytic systems with enhanced efficiency and selectivity could become accessible.

## Data availability
All data are available in the main text or the supplementary materials.

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

## Acknowledgements

We thank the Max–Planck Society (Max—Planck—Fellow Research Group 'Origins of Life', O.T.), Deutsche Forschungsgemeinschaft (DFG, German Research Foundation) Project—ID 521256690—TRR 392 (O.T.), Germany's Excellence Strategy, ORIGINS, EXC–2094—390783311 (O.T.), and the Volkswagen Stiftung, Initiating Molecular Life (O.T.) for funding.

## Author contributions

P.M., G.B. and O.T. conceived and designed the experiments. P.M. synthesised and characterized the compounds. P.M., G.B. and A.F.S. performed the kinetic measurements and in situ HRMS reaction profiling. P.M., G.B. and L.H. analysed the data. P.M. conducted and analysed the DHPLC experiments. L.H. and O.T. performed the reaction network analysis and kinetic analysis. P.M., G.B., A.F.S. and O.T. wrote the paper. All authors discussed the results and commented on the manuscript.

## Funding

## Competing interests

The authors declare no competing interests.
