## [Transparent Peer Review file · Nature Communications]

Mechanistic Analysis and Kinetic Profiling of Soai's Asymmetric Autocatalysis for Pyridyl and Pyrimidyl Substrates

Corresponding Author: Professor Oliver Trapp

Version 0:

Reviewer comments:

Reviewer #1

(Remarks to the Author)

The Soai reaction, i.e., 5-pyrimidyl alkanol acts as asymmetric autocatalyst in the enantioselective addition of diisopropylzinc to pyrimidine-5-carbaldehyde to afford more of itself, is unusual in which a chiral molecule self-replicates and extremely low (ca. 0.00005%) ee can be amplified to >99.5% ee during the three consecutive asymmetric autocatalysis. The reaction gives insights in the origins of homochirality of biomolecules although the molecule is not a biological one. Therefore, the Soai reaction has attracted much attention, and the mechanism and kinetics of the reaction have become one of the hot subjects of research in the world. However, the mechanism of the Soai reaction has not yet been fully understood. Professor Trapp and co-workers utilized in situ high-resolution mass spectrometry (in situ HRMS) and identified hemiacetal intermediate formed from isopropylzinc alkoxide of pyrimidyl (pyridyl) alkanol and pyrimidine-5-carbaldehyde (pyridine-3-carbaldehyde), and other combinations. The authors found that the hemiacetal structures are rather common in the Soai reaction. They clarified that the hemiacetals really exist in the reaction mixture and that hemiacetal-catalyzed pathway should exist by in situ HRMS. They also describe the kinetics involving hemiacetal by using HPLC and computer calculation. The authors describe that aldehyde is alkylated diastereoselectively on the chiral hemiacetal and that this could be the origin of amplification of ee (Fig. 2b). In the present manuscript, the substrates have been expanded to pyridyl alkanol system, and the intermediates (I6, I7, and I8) were newly found. Thus, in depth kinetic analysis was made possible. The experiments and the analysis are robust. This manuscript provides new insights into the mechanism of the Soai reaction. It will give an important hint for the future planning of an asymmetric autocatalysis. This reviewer strongly recommends the publication of this manuscript in Nature Communications after minor revisions.

(1) P.2, 1. Introduction, line 7.

“four reaction” should be “three reaction”

(2) P.2, 1. Introduction, line 2 from bottom.

To clarify, “alkyl zinc” should be “dialkyl zinc.”

(3) P.5, IIa In situ High Resolution.....

Second paragraph, line 2.

For clarification: Move structure number I3 to hemiacetal I3 resulting....

(4) P.6, IIa In situ High Resolution..... Line 6-7.

(5) For clarification. Move structure number I7 to intermediate complex I7.

(6) P.8, IIb Experimentally Derived Kinetics, line 5 from bottom.

Key number of ref. 4231 should be corrected.

(7) P.13, Fig. 4.

Spelling “ethinyl” should be “ethynyl.”

(8) Ref. 1 and 47 are the same.

Reviewer #2

(Remarks to the Author)

This paper focuses on the Soai reaction and its mechanism, which is still the subject of much debate in the literature. In this study, the authors made a mechanistic analysis with both pyridyl and pyrimidyl substrates. Using in situ high-resolution mass spectrometry, kinetic analysis and reaction profile simulations, they show that a catalytically active transient zinc

isopropyl hemiacetalate complex is involved in both substrates, which is an important discovery in determining the mechanism of this reaction. This conclusion calls into question recent work in the field.

The key elements supporting this discovery are in particular the identification of key intermediates (hemiacetals and its adducts with the reagents) during catalysis, the observation of a second-order kinetic with respect to the aldehyde derivative and the inclusion of a side-reaction when determining the latter, lifting contradictions from previous analyses in the literature. The data and the analyses are convincing and provide a remarkable re-investigation of the Soai's reaction. Due to the importance of the subject and the discovery described here, this work may be published. However, there are a number of points that need to be clarified before the publication can be accepted:

-A first condition for observing asymmetric autocatalytic amplification is the presence of a nonlinear effect. While it seems clear that a reservoir effect involving zinc alkoxide dimers is most likely the source of the NLE in the Soai reaction (as pointed out in their Chem. Eur. J. paper on that matter), the authors suggest the presence of diastereomers as the origin (abstract, p.5 and conclusion). However, the publication provides no evidence to support this hypothesis. Since the subject of this work is not the origin of NLE, this kind of statement confuses the message of the work. The authors should revise this part for clarity.

-p.4: two types of residues have been used (e.g., adamantyl and TMS). The authors should comment on why they chose these residues.

- This work calls into question previous work by the Denmark group, which suggested tetrameric zinc alkoxide as the active species using pyridinyl substrates. The authors need to put their results in the context of that previous work in a clear and unambiguous way. In particular, the following points should be addressed:

1) Do the in situ HRMS analyses here note the presence of tetrameric alkoxide zinc species, even in small quantities? The authors may discuss their results in context of tetramers previously observed for AdPym by DOSY NMR (Blackmond, Brown and co-workers, JACS 2010).

2) Is a tetramer model (possibly running in parallel to the hemiacetal-mediated mechanism) excluded by this work or not?

3) An important (and convincing) element here is the determination of a kinetic order of two with respect to the aldehyde. These results contradict those of Denmark, who observed an order of 0 (Nat. Chem. 2020). The authors should comment on that difference and where it originates from.

- Fig. 4 & the associated kinetic model: it is surprising to see that within the catalytic cycle there are only forward rate constants. This is scientifically not sound (broken microscopic reversibility) and even more so in dialkylzinc chemistry which is subject to fast equilibria. The authors should provide kinetic modelling that includes backward rate constants in the cycle, or bring compelling justification for their choice not to include them.

- Can the authors clarify the molecular structure of the hemiacetals? They are drawn in the schemes with their aromatic "arms" in a perpendicular fashion; in the DFT calculations on the C-C bond formation of their previous work (Chem. Eur. J 2020) the arms seem to be arranged in a parallel fashion.

Minor comment:

-Sup Fig 4.3.3, 4.3.5, 4.5.7: dark and light blue curves are inverted

-References: reference 1 should be Kagan's papers from 1986 et 1994 both published in JACS. The current reference 1 is a review that is already numbered as ref. 47 in the manuscript.

-Noyori and co-workers already investigated the matter of aldehyde reduction by dialkylzinc reagents, especially when using an excess of aldehyde (JACS 1989, p. 4028). This correlates with the observation discussed on p. 10 and should be cited there.

-References: there are typos in the numbering: 36, 37, 36, 39. I assume that 36 should be 38 ?

-References: the reference 37 should also include the original paper (i.e. Nat. Catal. 2020).

Reviewer #3

(Remarks to the Author)

Reviewer #4

(Remarks to the Author)

591690_0

Mechanistic Analysis and Kinetic Profiling of Soai's Asymmetric Autocatalysis for Pyridyl and Pyrimidyl Substrates.

The manuscript presents a thorough study of the Soai reaction i.e. the alkylation of pyrimidinyl aldehydes with diisopropyl zinc. A prominent feature of this system is that the produced chiral pyrimidine alkanol accelerates its own formation and promotes the prevalence of its own configuration. Therefore, the Soai reaction represents a unique laboratory demonstration of enantioselective autocatalysis. Since, its intimate mechanism is still controversial, any new investigation is welcomed. The purpose of the present manuscript is to combine experimental and theoretical approaches. It uses HRMS and kinetic analysis to elucidate this complex mechanism involving multiple reaction equilibria, transient intermediates and structural effects. The study was achieved by synthesizing 4 aldehyde derivatives with pyridine or pyrimidine aromatic cycle and trimethylsilyl- or adamantyl-ethynyl substituents. Although, the synthetic work has provided 11 new compounds, that HRMS measurements has allowed the identification of many reaction intermediates and that the modelling approach has been

improved by introducing a RS interaction, this manuscript appears to be a direct continuation of previous papers of the same group (refs 42 and 43). Therefore, it cannot be recommended for the broad readership of Nature Communications, but rather to a more specialized journal.

Before to be re-submitted, various points deserve to be improved and clarified.

The sentence: "As the rapid formation of homochiral (R,R)-2 and heterochiral (R,S)-2 dimers is the underlying phenomenon of the +NLE in the Soai reaction, this can be an explanation for a generally increased efficiency of the self-amplification of the pyrimidine autocatalytic systems" is not understandable.

Comment: There is a difference between the so-called NLE in asymmetric synthesis (Kagan, Noyori, etc.) and the spontaneous mirror-image symmetry breaking occurring in the Soai reaction. The Soai reaction is sensitive to minute cryptochiral additives, while asymmetric reactions are not, since they need a sufficient amount of chiral catalyst to operate. Running the Soai reaction in presence of an enantiomerically pure additive as it is carried-out in this manuscript, suppress all its ee amplification power. Looking at most of the kinetic records show that the final enantiomeric excess of the carbinols is often less than 99.9%. A reference about the cryptochiral amplification in the Soai reaction: Matsumoto et al., Angew. Chem. Int. Ed. 2016, 15246.

The sentences: "This distinct kinetic behavior of pyridine and pyrimidine autocatalytic systems displays the consistency of the structure-reactivity relationship with a mechanistic pathway involving a transient hemiacetal as the catalyst" and

"The concentration of the zinc hemiacetalate complex increases until an apex is reached at the inflection point of the s-shaped sigmoidal kinetic profile of the formed product alcohol and then eventually depletes"

deserve to be more balanced in the view of the following comments.

Comment: Looking at the figure 321 (supp. Info), it can be seen that the fastest alcohol evolution (around 5000s) occurs when the zinc hemiacetal is lower (around 0.4). On figure 322, the fastest alcohol evolution (around 500s) occurs when the hemiacetal is around 0.3 (not at its maximum). On figure 2c (main text), there is long inflexion point from 300 to 1250s during which the rate of alcohol formation is constant while the hemiacetal varies from 0.6 to 1, then to 0.3. Overall, while the data are noisy, the expected correlation between the rate of alcohol formation and the amount of hemiacetal is not so obvious. The fastest alcohol evolution does not occur when the hemiacetal is at its maximum. This is only on figure 323 where this correlation is perhaps visible. Therefore, the involvement of the hemiacetal is not demonstrated beyond a reasonable doubt and even more, since there is a paper from Gridnev et al. (BSCJ, 2015, 88,333) which concludes that the experimentally detected acetal species forms in the off-loop of the catalytic cycle and is not an important intermediate in the process of ee amplification. Few of the HRMS transients have been assumed to be involved into the reaction cycle. A brief mention of the study's limitation could be helpful.

Homo- vs heterodimerization of R- and S-zinc alcoholate has been assumed at $K3/K2 = 2$ (hetero more stable).

Comment: Such a value is not in agreement with Blackmond's claims (see ref. 18).

Other cosmetic and/or typographic improvements:

The layout of elements in Figure 1 was somewhat difficult to interpret. A visual reorganization might improve clarity at first glance.

On figure 4: (R)-1 = alcohol, while on figure 621 (supp.info): (R)-1 = zinc alcoholate.

To be rejected

Reviewer #5

(Remarks to the Author)

Version 1:

Reviewer comments:

Reviewer #2

(Remarks to the Author)

The authors have responded correctly to our comments in their resubmission. I therefore recommend publication of the manuscript as it stands now.

Minor comment: does the determination of reaction orders include an uncertainty value? If so, could you please include them in the corresponding table.

Reviewer #3

(Remarks to the Author)

Reviewer #4

(Remarks to the Author)

I agree that the Soai reaction mechanism is complex and can be interestingly debated.

However, looking at scheme 3 of reference: Chem. Eur. J. 26, 15871-15880 (2020) and figure 3 of reference: Front. Chem. 8, 1173 (2020), it can be seen that they are very similar to the figure 4 of this manuscript. Therefore, the expected originality of a Nature Comm. Paper is questionable.

Moreover, it's a pity that all the experiments shown on the S.I. (figures 4.2.3 to 4.2.11) use an enantiopure carbinol, thus suppressing the extraordinary ee amplification power of the Soai reaction.

This manuscript could be published but, in a more specialized journal.

To be rejected.

Reviewer #5

(Remarks to the Author)

List of Changes

Manuscript NCOMMS-25-17347-T

On behalf of all the authors, I would like to thank the competent reviewers for providing us with great feedback on our manuscript. We greatly appreciate all the helpful suggestions and valuable comments provided by the reviewers to improve the quality of the manuscript.

Point-by-Point responses to reviewers

REVIEWERS COMMENTS (black)

Our response (blue)

Reviewer #1 (Remarks to the Author):

The Soai reaction, i.e., 5-pyrimidyl alkanol acts as asymmetric autocatalyst in the enantioselective addition of diisopropylzinc to pyrimidine-5-carbaldehyde to afford more of itself, is unusual in which a chiral molecule self-replicates and extremely low (ca. 0.00005%) ee can be amplified to >99.5% ee during the three consecutive asymmetric autocatalysis. The reaction gives insights in the origins of homochirality of biomolecules although the molecule is not a biological one. Therefore, the Soai reaction has attracted much attention, and the mechanism and kinetics of the reaction have become one of the hot subjects of research in the world. However, the mechanism of the Soai reaction has not yet been fully understood. Professor Trapp and co-workers utilized in situ high-resolution mass spectrometry (in situ HRMS) and identified hemiacetal intermediate formed from isopropylzinc alkoxide of pyrimidyl (pyridyl) alkanol and pyrimidine-5-carbaldehyde (pyridine-3-carbaldehyde), and other combinations. The authors found that the hemiacetal structures are rather common in the Soai reaction. They clarified that the hemiacetals really exist in the reaction mixture and that hemiacetal-catalyzed pathway should exist by in situ HRMS. They also describe the kinetics involving hemiacetal by using HPLC and computer calculation. The authors describe that aldehyde is alkylated diastereoselectively on the chiral hemiacetal and that this could be the origin of amplification of ee (Fig. 2b). In the present manuscript, the substrates have been expanded to pyridyl alkanol system, and the intermediates (I6, I7, and I8) were newly found.

Thus, in depth kinetic analysis was made possible.

The experiments and the analysis are robust. This manuscript provides new insights into the mechanism of the Soai reaction. It will give an important hint for the future planning of an asymmetric autocatalysis. This reviewer strongly recommends the publication of this manuscript in Nature Communications after minor revisions.

We thank for the very positive feedback, the very thorough report and recommendations to improve our manuscript.

(1) P.2, 1. Introduction, line 7.

“four reaction” should be “three reaction”

(2) P.2, 1. Introduction, line 2 from bottom.

To clarify, “alkyl zinc” should be “dialkyl zinc.”

(3) P.5, IIa In situ High Resolution.....

Second paragraph, line 2.

For clarification: Move structure number I3 to hemiacetal I3 resulting....

(4) P.6, IIa In situ High Resolution..... Line 6-7.

(5) For clarification. Move structure number I7 to intermediate complex I7.

(6) P.8, IIb Experimentally Derived Kinetics, line 5 from bottom.

Key number of ref. 4231 should be corrected.

(7) P.13, Fig. 4.

Spelling “ethinyl” should be “ethynyl.”

(8) Ref. 1 and 47 are the same.

We appreciate these suggestions and corrections (1-8) for improvement, which we have incorporated into our manuscript.

Reviewer #2 (Remarks to the Author):

This paper focuses on the Soai reaction and its mechanism, which is still the subject of much debate in the literature. In this study, the authors made a mechanistic analysis with both pyridyl and pyrimidyl substrates. Using in situ high-resolution mass spectrometry, kinetic analysis and

reaction profile simulations, they show that a catalytically active transient zinc isopropyl hemiacetalate complex is involved in both substrates, which is an important discovery in determining the mechanism of this reaction. This conclusion calls into question recent work in the field.

The key elements supporting this discovery are in particular the identification of key intermediates (hemiacetals and its adducts with the reagents) during catalysis, the observation of a second-order kinetic with respect to the aldehyde derivative and the inclusion of a side-reaction when determining the latter, lifting contradictions from previous analyses in the literature. The data and the analyses are convincing and provide a remarkable re-investigation of the Soai's reaction. Due to the importance of the subject and the discovery described here, this work may be published.

We appreciate the comments of reviewer #2 very much. We have considered all points and addressed them in the following.

However, there are a number of points that need to be clarified before the publication can be accepted:

-A first condition for observing asymmetric autocatalytic amplification is the presence of a nonlinear effect. While it seems clear that a reservoir effect involving zinc alkoxide dimers is most likely the source of the NLE in the Soai reaction (as pointed out in their Chem. Eur. J. paper on that matter), the authors suggest the presence of diastereomers as the origin (abstract, p.5 and conclusion). However, the publication provides no evidence to support this hypothesis. Since the subject of this work is not the origin of NLE, this kind of statement confuses the message of the work. The authors should revise this part for clarity.

We appreciate this valuable feedback. Indeed, the reviewer is correct and we have rephrased the concerning parts to clarify that the NLE of the Soai reaction is, to the best of our knowledge, attributed to the diastereomeric dimers of the zinc alkoxide. The diastereomeric zinc hemiacetalate, which is formed upon addition of the chiral zinc alkoxide to a precursor, does not directly contribute to the NLE of the reaction but acts as an efficient catalyst for the asymmetric alkyl transfer.

-p.4: two types of residues have been used (e.g., adamantyl and TMS). The authors should comment on why they chose these residues.

The chosen residues for our studies, adamantyl and TMS, are reportedly good substituents on the alkyne and enable efficient autocatalytic behaviour. Furthermore, these residues complement our previous work on the tBuPym system. The results therefore allow conclusions on the influence of the residue on autocatalytic behaviour. We value this comment and have further addressed the substrate selection for our mechanistic investigations in the manuscript.

- This work calls into question previous work by the Denmark group, which suggested tetrameric zinc alkoxide as the active species using pyridinyl substrates. The authors need to put their results in the context of that previous work in a clear and unambiguous way. In particular, the following points should be addressed:

1) Do the in situ HRMS analyses here note the presence of tetrameric alkoxide zinc species, even in small quantities? The authors may discuss their results in context of tetramers previously observed for AdPym by DOSY NMR (Blackmond, Brown and co-workers, JACS 2010).

This is really an important point to address. In our In Situ HRMS we observed the presence of dimeric alkoxides, higher aggregates such as tetramers were not observed for any of the investigated systems. The identification of tetramers, for example by NMR by Blackmond and Denmark or by x-ray crystallography by Soai, have usually been a result of investigations under isolated conditions. We assume that tetrameric species are not present in significant amounts under reaction conditions, especially at early stages of the Soai reaction when the concentration of alkoxide is low.

2) Is a tetramer model (possibly running in parallel to the hemiacetal-mediated mechanism) excluded by this work or not?

During our research we were seeking to investigate the possibility of parallel autocatalytic cycles of hemiacetal and tetramer catalysis, however we did not find any experimental evidence for a tetramer cycle during any of our conducted experiments. The simulated and experimental kinetic

data were in good agreement with the hemiacetal cycle as the only proceeding pathway. The answer to this question would therefore be that our work excludes a tetramer mechanism running in parallel. We therefor modified our sentence in the manuscript to: “Higher aggregates of the zinc alkoxide, such as tetramers, were not observed detected under these reaction conditions for any of the four autocatalytic systems.”

3) An important (and convincing) element here is the determination of a kinetic order of two with respect to the aldehyde. These results contradict those of Denmark, who observed an order of 0 (Nat. Chem. 2020). The authors should comment on that difference and where it originates from.

Expanding on this point would indeed be informative for the readers of the manuscript. During our studies the rate dependency in aldehyde concentration was obvious and it seems rather unclear how this fundamental kinetic characteristic of the Soai reaction can be overlooked. The zeroth order rate dependency determined by Denmark and co-workers was obtained from a non-autocatalytic reaction in which the unsubstituted pyridine-3-carbaldehyhde was used as a substrate (Nat. Chem. 2020., SI, p. 105) and only the conversion of the aldehyde was monitored by in situ IR. However, as we have now shown it is necessary to not only account the aldehyde consumption into kinetic considerations but also the product formation and the reductive side reaction. Especially slow reacting substrates, such as the unsubstituted pyridine-3-carbaldehyhde, often show high amounts of reduction product which is obtained upon hydride transfer from the zinc alkylating agent. This reductive side reaction could therefore be the dominating pathway during the kinetic investigations of Denmark and coworkers which subsequently will give a zeroth order rate dependency in aldehyde concentration. We have included these considerations into our manuscript.

- Fig. 4 & the associated kinetic model: it is surprising to see that within the catalytic cycle there are only forward rate constants. This is scientifically not sound (broken microscopic reversibility) and even more so in dialkylzinc chemistry which is subject to fast equilibria. The authors should provide kinetic modelling that includes backward rate constants in the cycle, or bring compelling justification for their choice not to include them.

We have added an explanation for the simplification of our kinetic reaction model: “The microscopic reversibility was considered in the equilibria of the formation of the dimeric alcoxides and the hemiacetalate 5. Therefore, equilibria constants were introduced to determine the backward reaction rates in these steps. Considering the experimental observation that the intermediates in the autocatalytic cycle are formed at low concentrations and immediately consumed by the successive steps, we neglected equilibria and the backward reactions because of the high conversion rates in these steps. This assumption is also based on the finding that the reverse reaction of the final dimeric alkoxide is not noticeably altered in the presence of diisopropylzinc, which suggests, that the corresponding equilibria constants are high in the autocatalytic reaction cycle.”

In addition, it has to be pointed out that the addition of the backward reactions enormously expands the equations, which adds to the determination of the reaction rates in the system even more complexity.

- Can the authors clarify the molecular structure of the hemiacetals? They are drawn in the schemes with their aromatic “arms” in a perpendicular fashion; in the DFT calculations on the C-C bond formation of their previous work (Chem. Eur. J 2020) the arms seem to be arranged in a parallel fashion.

Thank you very much for addressing the depiction of the hemiacetals. Indeed, we have now corrected the structures according to the DFT calculations reported in our previous publication. All structures in the manuscript and in the SI are corrected, except for the structure in Figure 1, because it refers to a representation published. The problem was the overlap of the structures, the assignment of the configuration according to the CIP rules were not affected by this.

Minor comment:

-Sup Fig 4.3.3, 4.3.5, 4.5.7: dark and light blue curves are inverted

-References: reference 1 should be Kagan’s papers from 1986 et 1994 both published in JACS. The current reference 1 is a review that is already numbered as ref. 47 in the manuscript.

-Noyori and co-workers already investigated the matter of aldehyde reduction by dialkylzinc

reagents, especially when using an excess of aldehyde (JACS 1989, p. 4028). This correlates with the observation discussed on p. 10 and should be cited there.

-References: there are typos in the numbering: 36, 37, 36, 39. I assume that 36 should be 38 ?

-References: the reference 37 should also include the original paper (i.e. Nat. Catal. 2020).

We are thankful for the valuable suggestions for improvement by the reviewer and have incorporated the above-mentioned minor comments into our manuscript.

Reviewer #4 (Remarks to the Author):

591690_0

Mechanistic Analysis and Kinetic Profiling of Soai's Asymmetric Autocatalysis for Pyridyl and Pyrimidyl Substrates.

The manuscript presents a thorough study of the Soai reaction i.e. the alkylation of pyrimidinyl aldehydes with diisopropyl zinc. A prominent feature of this system is that the produced chiral pyrimidine alkanol accelerates its own formation and promotes the prevalence of its own configuration. Therefore, the Soai reaction represents a unique laboratory demonstration of enantioselective autocatalysis. Since, its intimate mechanism is still controversial, any new investigation is welcomed.

We thank the reviewer for these valuable comments, acknowledging the progress made in this field.

The purpose of the present manuscript is to combine experimental and theoretical approaches. It uses HRMS and kinetic analysis to elucidate this complex mechanism involving multiple reaction equilibria, transient intermediates and structural effects. The study was achieved by synthesizing 4 aldehyde derivatives with pyridine or pyrimidine aromatic cycle and trimethylsilyl- or adamantyl-ethynyl substituents. Although, the synthetic work has provided 11 new compounds, that HRMS measurements has allowed the identification of many reaction intermediates and that the modelling approach has been improved by introducing a RS interaction, this manuscript appears to be a direct continuation of previous papers of the same group (refs 42 and 43). Therefore, it cannot be recommended for the broad readership of Nature Communications, but rather to a more specialized journal.

We do not agree with the opinion of the reviewer, because it is not only an extension or application of previous knowledge to new substrates. In this work we can show unambiguously for the first time that the mechanism for the pyridyl and pyrimidyl substrates are very similar, however with very different reaction kinetics. This explains not only the differences in these two

systems, but we provide also quantitative results for the different substituents, here TMS and adamantly, which were also under debate in other contributions. In addition, we were able to identify and characterize the missing puzzle pieces in the mechanism, which corroborates the proposed autocatalytic cycle. This gain in knowledge is very important for the design of novel autocatalytic systems – which are of course not the topic of this research contribution.

Before to be re-submitted, various points deserve to be improved and clarified.

Thank you for the recommendation of resubmission, we have addressed all the points and comments made.

The sentence: “As the rapid formation of homochiral (R,R)-2 and heterochiral (R,S)-2 dimers is the underlying phenomenon of the +NLE in the Soai reaction, this can be an explanation for a generally increased efficiency of the self-amplification of the pyrimidine autocatalytic systems” is not understandable.

We have rephrased this sentence to clarify the observed self-amplification, according to the reported kinetic data: “As the rapid formation of homochiral (R,R)-2 and heterochiral (R,S)-2 dimers is the underlying prerequisite for the development of equilibria, which lead to a self-amplification in the Soai reaction. This can be an explanation for a generally increased efficiency of the pyrimidine autocatalytic systems.”

Comment: There is a difference between the so-called NLE in asymmetric synthesis (Kagan, Noyori, etc.) and the spontaneous mirror-image symmetry breaking occurring in the Soai reaction. The Soai reaction is sensitive to minute cryptochiral additives, while asymmetric reactions are not, since they need a sufficient amount of chiral catalyst to operate. Running the Soai reaction in presence of a enantiomerically pure additive as it is carried-out in this manuscript, suppress all its ee amplification power. Looking at most of the kinetic records show that the final enantiomeric excess of the carbinols is often less than 99.9%. A reference about the cryptochiral amplification in the Soai reaction: Matsumoto et al., *Angew. Chem. Int. Ed.* 2016, 15246.

Thank you for this comment and explanation. We completely agree. The mention reference was therefore already included in the original manuscript.

The sentences: “This distinct kinetic behavior of pyridine and pyrimidine autocatalytic systems displays the consistency of the structure-reactivity relationship with a mechanistic pathway involving a transient hemiacetal as the catalyst”

and

“The concentration of the zinc hemiacetalate complex increases until an apex is reached at the inflection point of the s-shaped sigmoidal kinetic profile of the formed product alcohol and then eventually depletes”

deserve to be more balanced in the view of the following comments.

Comment: Looking at the figure 321 (supp. Info), it can be seen that the fastest alcohol evolution (around 5000s) occurs when the zinc hemiacetal is lower (around 0.4). On figure 322, the fastest alcohol evolution (around 500s) occurs when the hemiacetal is around 0.3 (not at its maximum). On figure 2c (main text), there is long inflexion point from 300 to 1250s during which the rate of alcohol formation is constant while the hemiacetal varies from 0.6 to 1, then to 0.3. Overall, while the data are noisy, the expected correlation between the rate of alcohol formation and the amount of hemiacetal is not so obvious. The fastest alcohol evolution does not occur when the hemiacetal is at its maximum. This is only on figure 323 where this correlation is perhaps visible. Therefore, the involvement of the hemiacetal is not demonstrated beyond a reasonable doubt and even more, since there is a paper from Gridnev et al. (BSCJ, 2015, 88,333) which concludes that the experimentally detected acetal species forms in the off-loop of the catalytic cycle and is not an important intermediate in the process of ee amplification. Few of the HRMS transients have been assumed to be involved into the reaction cycle. A brief mention of the study’s limitation could be helpful.

We appreciate the attentive observations on the in situ HRMS measurements. Here, the reviewer has addressed the temporal evolvement of the hemiacetalate, the substrate aldehyde and the product alcohol. The data was obtained by injecting a running Soai reaction directly into an Orbitrap mass spectrometer, which is a difficult experimental task and several complications have

to be overcome, such as precipitation of zinc oxides in the ion source. Therefore, the quantification of reaction intermediates can be somewhat noisy however the general correlation of hemiacetal abundance and alcohol formation rate is observable. Similar observations were also made by Blackmond by NMR tracking (*Angew. Chem.* 2012, 124, 1-5). Furthermore, the reviewer should consider that temporal tracking was only one result of the in situ HRMS measurements which support the hemiacetal involvement, as we found several intermediate complexes of the hemiacetal which constitute the alkylation step of the autocatalytic cycle.

The experimental data reported are sound and are the basis for the mechanistic findings. We know very well the paper by Gridnev (Gridnev, I. D. & Vorobiev, A. K. On the origin and structure of the recently observed acetal in the Soai reaction. *Bull. Chem. Soc. Jpn.* **88**, 333-340 (2015), which opposes also the findings by Blackmond and Brown. However there are a few mistakes in this paper, which complicate the justifications made by the reviewer. For example in Figure 4 a phenol is depicted in the equilibrium of the TS instead of the expected Soai alcohol. Since the DFT calculations were performed on a quite low level (B3LYP/ 6-31G*), it is difficult to conclude that hemiacetals are not involved. Our experimental findings disprove these theoretical assumptions.

Homo- vs heterodimerization of R- and S-zinc alcoholate has been assumed at $K_3/K_2 = 2$ (hetero more stable).

Comment: Such a value is not in agreement with Blackmond 's claims (see ref. 18).

This comment is not justified and wrong.

In the mentioned reference from Blackmond (ref. 18: *JACS* 2006, 125, 8978-8979, and in *Tetrahedron Asymmetry* 2006, 17, 584-589) it is stated: $(K_{\text{hetero}}/K_{\text{homo}})^2 = 4$ which is equal to $K_{\text{hetero}}/K_{\text{homo}} = 2$. See also Scheme 2 in Buono, F. G., Iwamura, H. & Blackmond, D. G. Physical and chemical rationalization for asymmetric amplification in autocatalytic reactions. *Angew. Chem. Int. Ed.* 43, 2099-2103 (2004).

Other cosmetic and/or typographic improvements:

The layout of elements in Figure 1 was somewhat difficult to interpret. A visual reorganization might improve clarity at first glance.

On figure 4: (R)-1 = alcohol, while on figure 621 (supp.info): (R)-1 = zinc alcoholate.

We have added some more space in figure 1 between a,b and c for a better visual separation of the elements.

List of Changes

Manuscript NCOMMS-25-17347-T

On behalf of all the authors, I would like to thank the competent reviewers for providing us with great feedback on our manuscript. We greatly appreciate all the helpful suggestions and valuable comments provided by the reviewers to improve the quality of the manuscript.

Point-by-Point responses to reviewers

REVIEWERS COMMENTS (black)

Our response (blue)

Reviewer #2 (Remarks to the Author):

The authors have responded correctly to our comments in their resubmission. I therefore recommend publication of the manuscript as it stands now.

Thank you very much for the very kind comment.

Minor comment: does the determination of reaction orders include an uncertainty value? If so, could you please include them in the corresponding table.

Thank you very much for this recommendation. We added the deviations to the values in table 1.

Reviewer #3 (Remarks to the Author):

No changes were requested.

Reviewer #4 (Remarks to the Author):

I agree that the Soai reaction mechanism is complex and can be interestingly debated.

However, looking at scheme 3 of reference: Chem. Eur. J. 26, 15871-15880 (2020) and figure 3 of reference: Front. Chem. 8, 1173 (2020), it can be seen that they are very similar to the figure

4 of this manuscript. Therefore, the expected originality of a Nature Comm. Paper is questionable.

The figures are definitely different and there are also changes in the depicted structures, which reflect our experimental findings. Furthermore, in the present manuscript different substrates were investigated and the absolute stereochemical assignment set. We do not agree with the reviewer.

Moreover, it's a pity that all the experiments shown on the S.I. (figures 4.2.3 to 4.2.11) use an enantiopure carbinol, thus suppressing the extraordinary ee amplification power of the Soai reaction.

The investigation of the amplification power of the Soai reaction was not the aim of this study, only the mechanism. Nevertheless, we would like to thank you for this suggestion. We will carry out such experiments and apply our kinetic model directly to them.

This manuscript could be published but, in a more specialized journal.

To be rejected.

We are not commenting this personal opinion.